# The Role of Ultrasound in the Diagnosis of Pulmonary Infection Caused by Intracellular, Fungal Pathogens and Mycobacteria: A Systematic Review

**DOI:** 10.3390/diagnostics13091612

**Published:** 2023-05-02

**Authors:** Mariaclaudia Meli, Lucia Spicuzza, Mattia Comella, Milena La Spina, Gian Luca Trobia, Giuseppe Fabio Parisi, Andrea Di Cataldo, Giovanna Russo

**Affiliations:** 1Pediatric Hematology and Oncology Unit, Department of Clinical and Experimental Medicine, University of Catania, 95123 Catania, Italy; mclaudiameli@gmail.com (M.M.); mattiacomella@gmail.com (M.C.); mlaspina@unict.it (M.L.S.); adicata@unict.it (A.D.C.); 2Pulmology Unit, Department of Clinical and Experimental Medicine, University of Catania, 95123 Catania, Italy; lucia.spicuzza@unict.it; 3Pediatrics and Pediatric Emergency Room, Cannizzaro Emergency Hospital, 95126 Catania, Italy; trobia@tin.it; 4Pediatric Pulmology Unit, Department of Clinical and Experimental Medicine, University of Catania, 95123 Catania, Italy; giuseppe.parisi@policlinico.unict.it

**Keywords:** lung ultrasound, pneumonia, pulmonary infection

## Abstract

Background: Lung ultrasound (LUS) is a widely available technique allowing rapid bedside detection of different respiratory disorders. Its reliability in the diagnosis of community-acquired lung infection has been confirmed. However, its usefulness in identifying infections caused by specific and less common pathogens (e.g., in immunocompromised patients) is still uncertain. Methods: This systematic review aimed to explore the most common LUS patterns in infections caused by intracellular, fungal pathogens or mycobacteria. Results: We included 17 studies, reporting a total of 274 patients with *M. pneumoniae*, 30 with fungal infection and 213 with pulmonary tuberculosis (TB). Most of the studies on *M. pneumoniae* in children found a specific LUS pattern, mainly consolidated areas associated with diffuse B lines. The typical LUS pattern in TB consisted of consolidation and small subpleural nodes. Only one study on fungal disease reported LUS specific patterns (e.g., indicating “halo sign” or “reverse halo sign”). Conclusions: Considering the preliminary data, LUS appears to be a promising point-of-care tool, showing patterns of atypical pneumonia and TB which seem different from patterns characterizing common bacterial infection. The role of LUS in the diagnosis of fungal disease is still at an early stage of exploration. Large trials to investigate sonography in these lung infections are granted.

## 1. Introduction

Since its first introduction in clinical practice, lung ultrasound (LUS) has been acknowledged as a potential first-line imaging modality to recognize common lung pathology [1,2,3]. Among a number of acute conditions, the accuracy and reliability of LUS for the diagnosis of community acquired pneumonia (CAP), has been explored with promising results in both adults and children [4,5,6]. There is now evidence that LUS may have greater sensitivity, similar specificity, and better inter-operator reliability in the diagnosis of pneumonia when compared with standard chest X-ray (CXR) [4,7,8,9,10]. Moreover, LUS has the advantage of being free of ionizing radiation, has lower cost and easier bedside availability than CXR and is subject to fewer regulatory requirements. This makes the technique particularly appealing as a point-of-care tool in the acute clinical setting, especially for children.

Indeed, although LUS has been under evaluation for over a decade, the important role played during the recent COVID-19 pandemic, has greatly increased the interest of clinicians from multiple disciplines toward this technique and many efforts have been done to standardize its use [11].

It is a paradigm that, to be detected from LUS, a parenchymal lesion must be close enough to the pleura. In patients with typical bacterial pneumonia, the parenchymal consolidation usually appears as a sub-pleural hypo-echoic area associated with hyper-echoic dynamic spots called “air bronchograms”, representing air-filled bronchi within the density of surrounding alveoli [12,13]. The pleural line above the consolidation is less echogenic or even non-visible. At the rear, the presence of compact vertical artifacts, called B lines, is a frequent expression of wall reinforcements typically produced by areas with fluid content [14,15].

Differently from bacterial pneumonia, interstitial pneumonia (e.g., viral) sonographycally appears as a pattern characterized by three or more B lines (vertical hyperechoic reverberations) in the same scansion between two ribs, either isolated or confluent. These features are similar to those described in the interstitial-alveolar syndrome and may form, in more severe cases, a “white lung pattern” [16].

As mentioned, a large number of studies have shown the accuracy of LUS in the diagnosis of the most common lung infections affecting children and adults, mainly bacterial pneumonia [4,10,17]. Conversely, few studies have explored the usefulness of LUS for the detection of less common lung infections such as those caused by atypical and fungal pathogens or mycobacteria.

Atypical pneumonia caused by intracellular pathogens such as Mycoplasma Pneumoniae (*M. pneumoniae*) and Chlamydia Pneumoniae (*C. pneumoniae*) distinctively cause lung interstitial involvement and are insensitive to common antibiotics used for the treatment of bacterial pneumonia [18]. Lung infections caused by mycobacteria, mainly *Mycobacterium tuberculosis* or fungal pathogens are rare in immunocompetent patients, but increasingly diagnosed in immunocompromised patients [19]. In addition, TB remains highly prevalent in low- and middle-income countries [20]. When a lung infection is suspected in immunocompromised patients, it is crucial to reach an early diagnosis in order to start promptly a specific treatment. CXR and computerized tomography (CT) are often required in febrile patients with cancer to make diagnosis. However, CXR, especially during neutropenia, is hampered by a low specificity and hardly differentiates bacteria from a non-bacterial pneumonia, making essential a chest CT scan that has both higher sensitivity and specificity [21]. The role of lung ultrasound as a point-of-care tool may therefore be decisive in the diagnosis and follow-up of these infections, particularly in children and severely ill patients.

This systematic review aimed to explore the most common findings reported at lung ultrasound in patients with infection caused by intracellular, fungal pathogens or mycobacteria, and to discuss a putative role for this technique in the diagnosis of these lung infections.

## 2. Materials and Methods

A search strategy was developed in order to recognize the most significant literature on the topic. An exhaustive search was performed on the main scientific libraries including PubMed, Embase, Google Scholar and Cochrane. We used the following keywords: lung/thoracic ultrasound/ultrasonography, atypical pneumonia, *Mycoplasma pneumoniae*, *Chlamydia pneumoniae*, fungal pneumonia, fungal invasive disease, lung Aspergillosis, tuberculosis/mycobacterium. A combination of MeSH and associated terms with other methodological terms (Mycoplasma or Chlamydia or atypical pneumonia or pneumonitis and LUS or lung or thoracic and ultrasound or ultrasonography; fungal or fungal invasive or Aspergillosis and pneumonia or pneumonitis and LUS or lung or thoracic and ultrasound or ultrasonography; Tuberculosis or Mycobacterium and pneumonia or pneumonitis and LUS or lung or thoracic and ultrasound or ultrasonography).

### Study Selection and Data Extraction

The search was limited to articles published in English peer-reviewed journals between January 1990 and January 2023. We included original studies, retrospective or prospective, case reports and case series reporting detailed lung ultrasound findings for the diagnosis or follow-up of pulmonary infection by atypical, fungal pathogens and mycobacteria. We included studies on adults, children and infants reporting a microbiologically confirmed lung infection. We excluded: (1) studies reporting data from endoscopic ultrasound; (2) studies reporting “suspected infection” without clear microbiology data; (3) study in which the prevalence of any ultrasound finding in the population was not well specified; (4) review articles and meta-analyses; (5) qualitative studies; (6) not English articles.

For data processing, the document management tool Mendeley and the program Microsoft Excel were used. The cumulative selection of articles was evaluated and screened independently by two researchers. In case of disagreement between investigators, a third investigator was involved. To select eligible studies for full text review, we used an evidence-based algorithmic approach, Preferred Reporting Items for Systematic Reviews and Meta-Analyses (PRISMA) (Figure 1) [22].

For each study we collected the following information: (1) author and year of publication; (2) sample size; age; (3) specific pathogen; (4) LUS findings.

## 3. Results

### 3.1. Study Selection and LUS Definitions

The initial search included a total of 695 studies. Duplicates found in different databases were excluded. From the remaining 550 we excluded 431 articles for being review, not in English or qualitative studies. Of 119 studies assessed for eligibility, only 17 articles were useful for the purpose of this review. These included four studies on atypical pneumonia, seven on fungal infection and six on pulmonary tuberculosis. Although sometimes a different terminology was used in the studies to describe lung patterns, definitions of the main signs observed from LUS are shown in Table 1.

### 3.2. Primary Results

#### 3.2.1. Intracellular Pathogens Lung Infection

All eligible studies on pneumonia caused by intracellular pathogens focused on *M. pneumoniae* except one study also including some cases of *C. pneumoniae*. As this infection is more common at a pediatric age, not surprisingly all studies were performed in children with an age range from 2 months to 15 years (Table 2). A total of four studies, two prospective and two retrospective were analyzed, including a total of 274 patients with a diagnosis of atypical pneumonia confirmed by microbiology and conventional imaging.

Overall, the sonographic patterns described were consistent among studies (Figure 2, Figure 3 and Figure 4). The most common LUS feature was consolidation reported in 84–100% of the patients (Table 2). Of the two studies analyzing the consolidation’s dimension, one reported most commonly a dimension < 1.5 cm, while another described largest consolidated areas up to 4 cm (Figure 2). Only in a few cases of severe atypical pneumonia did the consolidation reach the dimension of 6 cm. Atelectasis, characterized by a hyperechoic static air bronchograms, was also common (60%). In most of the cases consolidation was associated with subpleural effusions (Figure 3) and a diffuse interstitial pattern. B lines were in fact extremely common, being present in 85% of all cases (Figure 4). Although both scattered and confluent B lines were described, in most cases, the B lines were confluent and so dense in more severe disease that they formed the so-called “white lung”.

#### 3.2.2. Fungal Lung infection

Studies on pneumonia caused by fungal pathogens, also known as invasive fungal disease, were rare and extremely heterogeneous in terms of age range, disease manifestation, patients’ comorbidities (e.g., immune status) and specific pathogen (Table 3).

A total of seven studies were analyzed which included a total of 30 patients. These studies were mainly case reports describing between one and four patients. The three largest studies available included, respectively, 10 adults, 6 children (6–11 years) and 7 premature infants (Table 3). In most of the cases lung fungal disease was caused by Aspergillus, five cases of Candida Albicans and two cases of rare Mucormycosis were also described.

Among 10 cases of invasive fungal diseases, in patients treated with allogenic hematopoietic stem cell transplantation, the most common LUS feature was sub-pleural consolidation with an inhomogeneous echotexture and indistinct margins, usually bilateral (Figure 5). Atelectasis and pleural effusion were present in 40% of the patients, whereas B lines, expression of an interstitial pattern, were extremely common (80%) (Figure 6, Table 3).

Cavitations, a common feature of pulmonary Aspergillosis observed in lung CT scan, were reported in all four adults (age 49–79 years) belonging to a group with invasive aspergillosis and in two out of five children, aged 7–11 years, with Aspergillosis and Mucormycosis (Figure 7, Table 3).

In a group of seven premature newborns the sonographic pattern of lung invasive fungal disease was typical, characterized by bilateral lung consolidation with air bronchogram and irregular boundaries (Table 3). Different patterns of B lines, reflecting a different degree of lung edema, were observed in all non-consolidated areas.

In a group of older children (5–11 years), more typical lesions were described, including hypo-echoic nodules with hyperechoic rims or hyper-echoic nodules with hypo/anechoic rims (Figure 8, Figure 9 and Figure 10). In a few cases, the “fungus ball” was evidenced as a hyper-echoic area in the center of the main lesion (Figure 8).

#### 3.2.3. *Mycobacterium tuberculosis* Lung Infection

We selected six studies, including a total of 213 patients affected by pulmonary TB. Of these, one study included only children (40 cases, with a mean age of 2 years old) and one study specifically selected 10 adults with miliary TB (Table 4). The studies were performed either in Western countries including patients at high risk for TB or in countries that have a high risk of TB.

Sonographic consolidations, often multiples and mainly apical, where commonly described in both children (55%) and adults (from 77% to 80%), with the exception of cases of miliary TB. None of the studies, except a single case report, reported the occurrence of atelectasis in the context of consolidated areas.

In adult TB, the most common and peculiar findings were circular or ellipsoidal hypo-echoic sub-pleural lesion, generally <1.5 cm, defined as “sub-pleural nodes”. These were reported in up to 90% of adults, mainly in the superior quadrants of the lung (Table 4, Figure 11 and Figure 12).

Pleural effusion was most commonly found in children (30%) than in adults (on average 15% of the cases). However, pleural irregularities were common findings in all groups. Sonographic cavitation was absent in children and was reported in a small number of adults (5 to 30%) (Figure 13 and Table 4).

In 10 adults with miliary TB, a quite precise sonographic pattern was shown, invariably characterized by B lines and echogenic bright “granular” artifacts in the sub-pleural areas, defined as “sub-pleural granularities”.

## 4. Discussion

### 4.1. General Considerations

Over the last decade, a considerable amount of work has been performed to define the role of LUS as point-of-care diagnostic tool [46,47,48,49,50,51,52,53,54,55]. While most of the studies focused on CAP, with pathogens rarely microbiologically distinguished, LUS has also been proposed as a tool for etiological diagnosis [33]. This systematic revision explored cases of lung infection caused by pathogens which are rare in the immunocompetent population (Mycobacteria and fungi) or by intracellular pathogens. These infections are common in children but are mainly included in the group of community acquired infections, without any distinction from the most common etiologies such as *S. pneumoniae*. We found a limited number of studies, particularly for fungal infection. These studies presented several limitations including mainly a small sample size and heterogeneous definitions of lung sonographic patterns. This is not surprising, as LUS is still an emerging technique and is currently in the process of standardization. For this reason, the studies were difficult to compare and the study populations difficult to merge. However, we found that some interesting and specific LUS patterns have been described in association with these infections and can represent a good starting point for further investigations.

### 4.2. Intracellular Pathogens Lung Infection

As expected, data on the accuracy of LUS in atypical pneumonia focused on children, and indeed the role of this technique in pediatrics may be crucial, given the extent they are free of ionizing radiation [56,57,58,59,60,61,62,63,64,65,66,67]. A recent meta-analysis showed a 96% sensitivity and 93% specificity of LUS in detecting pneumonia in children [68]. The peculiar feature of atypical, as well as of viral pneumonia, is the involvement of lung interstitium with characteristic edema and inflammatory cellular infiltrate. These pathological changes correlate with specific radiographic features such as ground-glass and reticular/nodular patterns [69].

A mentioned above, the identification of a lung interstitial pattern linked to non-bacterial infection through the detection of B lines from LUS has been widely reported during the observation of COVID-19-related pneumonias [70].

Indeed, we found that the presence of an interstitial pattern, characterized by scattered or most commonly confluent B lines is a main feature in children with atypical pneumonia in all age groups [31,32,33,34,37,71,72,73].

Most of the studies that we examined aimed to correlate LUS and traditional imaging findings and to assess the possibility to rapidly differentiate the etiology of lung infection by bedside LUS.

In one of the largest prospective studies, Buonsenso et al. found that in children with pneumonia, LUS findings were more helpful than clinical presentation, laboratory data and CXR in distinguishing bacterial, viral or atypical pneumonia [33]. They showed that multiple consolidations were related to viral and atypical pneumonia, whereas larger and solitary consolidations with bacterial pneumonia. Moreover, deep air bronchogram was more typical of bacterial and viral pneumonia, whereas a superficial air bronchogram was almost always present in atypical pneumonia. B lines were more common in atypical/viral pneumonia, generally diffused to the lung, while in bacterial pneumonia B lines were mainly located in continuity with the solid mass. It is noteworthy that the same group has previously shown that LUS pattern recognition at diagnosis may help the predict early antibiotic response better than clinical and laboratory data [72].

The relevance of the dimension of the consolidation and distribution of B lines has been suggested earlier by studies on CAP, although these could not be included in this review since the exact prevalence of the findings was not reported. Data from Buonsenso et al. were in agreement with earlier data from Berce and co-workers. The data showed that in children with bacterial pneumonia consolidations, B lines were commonly solitary, larger, and unilateral compared to those with viral/atypical pneumonia. In bacterial pneumonia, B lines were in proximity to the consolidation, whereas in viral/atypical pneumonia, they were diffuse [17].

Interestingly, Iorio and co-workers describe some small sub-pleural consolidations in children with CAP and consider these “satellites” of the main consolidation, when they are located in the lower or upper anterior district of the contralateral lung [73]. They speculate that this phenomenon could be linked to the lymphatic drainage, being peculiar of atypical pneumonia in which the involvement of lymphatic network is part of the pathogenesis of the interstitial disease [73].

The patterns described so far have been further confirmed by Liu and co-workers in the a large and most recent study [34,37]. In all children with atypical pneumonia, an interstitial pattern was reported, which in some cases, formed the “white lung” pattern. Interestingly, these authors measured the ratio of consolidation size/body surface area and suggested that the dimension of the consolidation may depend on the age of the child [34,37]. They also found that the presence of pleural effusion, which can be detected even in a tiny quantity by LUS is a negative prognostic factor in pediatric atypical pneumonia [34,37].

The predictive value of LUS findings has also been explored by Li et al. [31]. They found that the evaluation of consolidation and atelectasis through LUS may predict the effect of bronchoalveolar lavage in the treatment of children with severe *M. pneumoniae* infection [31].

However, it has to be mentioned that other studies failed to find specific LUS findings for atypical pneumonia. As an example, Tripaldi et al. describe large consolidations in both bacterial and atypical CAP in 4-year-old children and state that the interstitial pattern is not so common in atypical pneumonia [32].

### 4.3. Fungal Lung Infection

Fungal pulmonary infection, also known as invasive fungal infection, encompasses a broad spectrum of conditions affecting patients who are immunocompromised for different reasons (e.g., chemotherapy, organ transplantation, blood cancer) [71,74,75,76]. These pathogens are a major threat as they are becoming increasingly common and resistant to treatment, causing high mortality. Although a number of fungal pathogens may infect the lung, infection by different species of Aspergillus is the most common [77,78,79,80].

The gold standard imaging for invasive fungal infection is lung CT showing a large variety of lesions including areas of consolidation, cavitation, abscess, nodule or infarction associated with the angio-invasive nature of the fungal pathogens [81,82,83,84,85]. In this wide range of lesions some signs at lung CT can be considered more specific, including: (1) “halo sign”, a crescent or complete ring of ground-glass opacity surrounding a focal rounded area of consolidation; (2) “reverse-halo” sign, a focal rounded area of ground-glass opacity surrounded by a crescent/complete ring of consolidation [81]. Although these signs may be present in other bacterial or viral infections, they are strongly suggestive of fungal infection in immunocompromised patients [86,87,88,89].

In many cases, these lesions are located in close proximity to the pleura, leading to speculation about the possible role of LUS in their detection. In fact, dealing with patients that are generally severely ill, and given that laboratory data are slow to obtain, the use of a point-of-care diagnostic tool could be crucial for a prompt management of the disease. In addition, it should also be considered that CRX has a low specificity for these infections [21]. Unfortunately, the literature on the topic is scarce. In addition, as lung CT scan may present a variety of different patterns, it might be more difficult to detect these patterns on LUS.

In one of the first studies on LUS-guided fine-needle aspiration biopsies in adults with pulmonary Aspergillosis, abscesses were described as both round, hypo-echoic areas with irregular margins and as difficult to differentiate from bacterial abscesses. However, with the help of LUS biopsies, they were diagnosed in three out of four patients [38].

Perhaps, the most representative study on the topic is the report by Alamdaran et al., describing ten children with hematological cancer and fungal lung infection (mainly Aspergillosis) all undergoing CT scan and LUS [36]. In the CT scan, the most common findings were nodules with halo-sign or reverse halo sign or crescent sign, wedge-shaped consolidation and cavitation. In the LUS, these patterns had a peculiar appearance as either hyper-echoic nodules with hypo-anecoic rims or hypo-echoic nodules with hyperechoic rims. The presence of a mycetoma in an existent cavitation is particular in chronic conditions. In the CT scan, it appears as a crescent sign, airspace between the mycetoma and the cavitation. The mycetoma can be detected by LUS and appears as a central hyperechoic roundish area with air extension to the peripheral anechoic rim, often described as the air crescent sign or cavitation [36,39]. In another cohorts of adults with invasive fungal disease, after hematopoietic stem cell transplantation, LUS showed a good sensitivity compared to CT for the detection of fungal infection, showing hypo-echoic areas with positive air bronchogram [40].

It has been suggested that the evaluation of the vascular component of the lesion with color-coded Doppler may help to discriminate a fungal pneumonia from other causes of consolidation [7,35,39].

Finally, it is worth mentioning a brief report showing the presence of consolidations with air bronchogram and B lines from LUS in premature newborn infants with fungal infection [37]. If confirmed, the latest data might be of great relevance given the difficulty diagnosing early onset pneumonia in newborn infants via traditional imaging.

### 4.4. Mycobacterium tuberculosis Infection

TB is still a major of cause of death worldwide and to achieve a global reduction of the disease, a prompt diagnosis and treatment initiation is fundamental [20,90,91,92]. The diagnosis of active TB is based on microbiology and traditional radiology; however, facilities for diagnosis are often lacking in countries with low resources where the prevalence of TB is significantly high. In this context, the use of ultrasonography as an affordable point-of-care diagnostic tool is extremely attractive. Indeed, given that LUS is becoming increasingly popular its use for suspected TB cases, it has been explored by a number of studies [45,62,93,94,95].

The three largest studies on adults TB included in this review showed that the most common LUS features were sub-pleural nodules, < 1 cm, often multiple and diffuse, and areas of consolidation, generally on apical or middle fields. In a recent systematic review in adult TB, it was shown that these sub-pleural nodules and lung consolidations were the LUS findings with the highest sensitivities, ranging from 72 to 100% and 46% to 80%, respectively [62].

According to one study, although sub-pleural consolidations may be present in other conditions, in TB, these may appear patchier and more irregular and contiguous with pleura [44].

However, in some cases, the consolidation in LUS has been described as indistinguishable from bacterial pneumonia [29].

The specificity of LUS findings has been reported only by one study showing that in the presence of suggestive symptoms, the combination of apical consolidations and sub-pleural oval or round nodules, can reach a specificity of 96% [43]. There is agreement among authors that for TB, LUS has a poor ability to screen radiographically identified cavities [29,41,43].

Although differences exist among studies, considering that the sensitivity of CXR for TB has been estimated around 87%, it is reasonable to expect that LUS, with a sensitivity ranging from 72% to 100%, has the potential to become a valid alternative point-of-care diagnostic tool. A recent study further supports this consideration, showing that in a cohort of 82 patients with suspected TB, the overall sensitivity was 80% for LUS and 81% for CXR [96]. It is also noteworthy that in some patients with TB, pleural effusion is more commonly evidenced by LUS than a CT scan [97]. One limitation of using LUS is that in many cases, TB lesions are localized in the posterior-superior regions of the lung, which may not be visible to LUS due to the presence of the scapula [96].

One of the first studies on LUS in TB explored the features of miliary manifestation, which is characterized by tiny (1–2 mm) diffuse granulomatous lesions appearing in CXR as multiple small opacities diffuse in all lung zones [30,98]. The recent interest toward this “old” disease has been raised due to the common occurrence of this miliary form in patients with HIV or in patients undergoing immunosuppressive therapy [98]. Sonographic findings in this form of TB consist invariably in an interstitial pattern typically present in all lung zones together with subpleural granularities [30]. While the interstitial pattern in immunocompromised patients may be shared by the rare Pneumocystis Jirovecii or Cytomegalovirus infections, sub-pleural granular changes may add specificity [30].

Finally, little can be concluded on the role of LUS in pediatric TB, as only one study was published showing that consolidation was commonly associated with pleural effusion and enlarged mediastinal lymph nodes [42]. Our review focused on identifying parenchymal and pleural patterns using sonography. However, in children with TB, there has been more exploration into sonographic detection of enlarged mediastinal lymph nodes, with the aim of performing mediastinal biopsies [99,100].

## 5. Conclusions

Lung sonography is a rapid, widely available, free-of-adverse effects and point-of-care diagnostic tool. LUS has been considered equal or even superior to chest X-ray for the detection of pneumonia in both adults and children. Although chest X-ray remains unsurpassed for the detection of central lesions, sonography can detect peripheral lesions, even small ones that can escape a chest X-ray. The use of LUS for less common lung infections has raised interest, although the literature is still insufficient to draw a definitive conclusion. In addition, available studies present acute limitations including (1) low number of patients, (2) methodological limitations, (3) poor standardization of the technique, and (4) a lack of common definitions for LUS findings. However, aside from these limitations, some aspects have emerged from this review that deserve consideration.

The role of ultrasonography has been explored in children with atypical pneumonia caused by M.pneumoniae, and certainly some specific patterns have been described. If these were to be confirmed by larger trial, it may position LUS as a point-of-care tool for distinguishing the etiology of community-acquired pneumonia and predicting outcomes and responses to therapy. In atypical pneumonia, consolidations where described in 84–100% of cases; in most of these cases, consolidation was associated with a diffuse interstitial pattern characterized by scattered or most frequently confluent B lines. Studies focusing on the role of LUS in the diagnosis of pulmonary TB have shown a good sensitivity of this technique compared with standard imaging. The most common LUS features were sub-pleural nodules, <1 cm, often multiple and diffuse, and consolidations, often multiples and mainly apical. Although scant data are available, it is common opinion that further investigation is worthy in this field. In fact, LUS might be a more easily affordable diagnostic tool in low-income countries where traditional imaging is not promptly available.

Sonographic imaging of lung invasive fungal infection has received less attention, and several reports have presented inconsistent data. The most common LUS feature was sub-pleural consolidation with an inhomogeneous echotexture and indistinct margins, which were usually bilateral. In cases of diffuse Aspergillosis infection, cavitations were also detected by LUS. However, given the poor outcomes and mortality associated with this disease, particularly in patients with malignancies, a bedside tool for the follow-up and treatment strategy is important. Therefore, large trials on the use of LUS for the differentiation between bacterial and fungal pneumonia are needed.

## Figures and Tables

**Figure 1 diagnostics-13-01612-f001:**
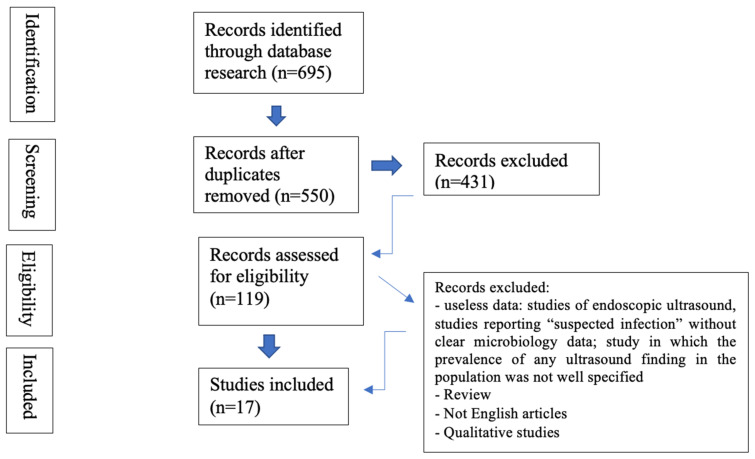
PRISMA flow-chart [22].

**Figure 2 diagnostics-13-01612-f002:**
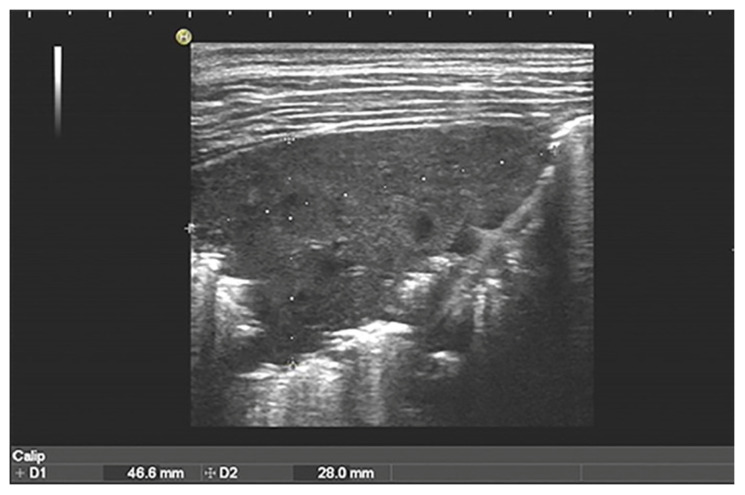
Cosolidation and air bronchogram in *Mycoplasma pneumonia* [31].

**Figure 3 diagnostics-13-01612-f003:**
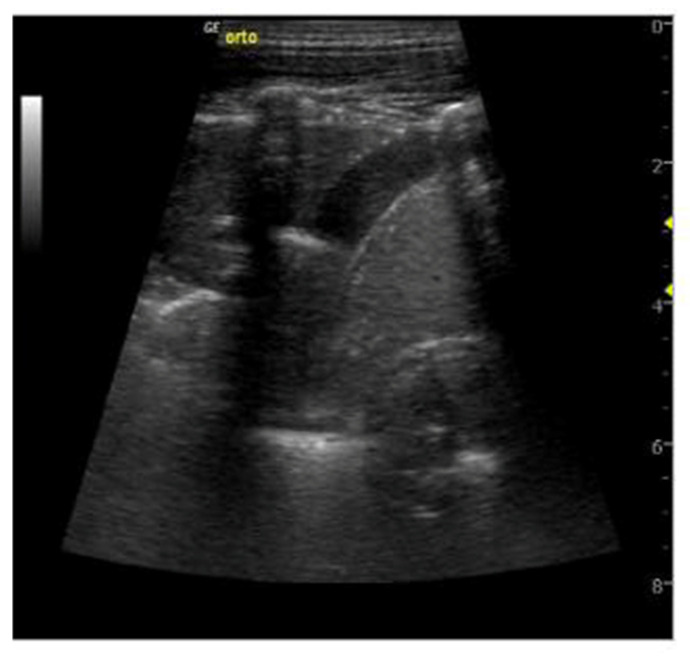
Subpleural effusions in *Mycoplasma pneumonia* [32].

**Figure 4 diagnostics-13-01612-f004:**
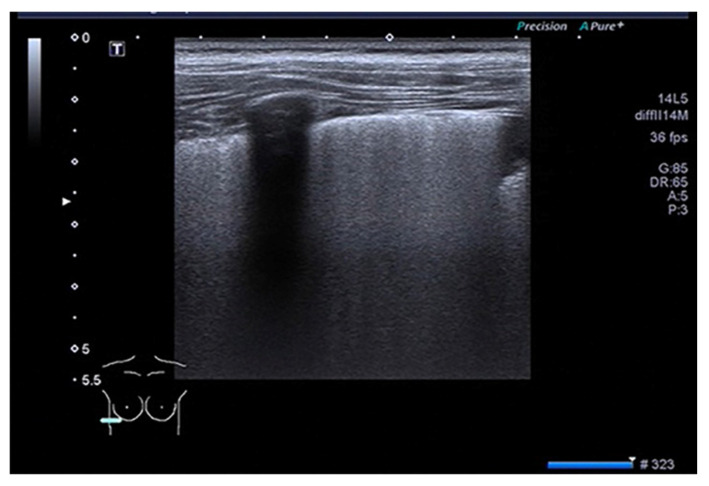
B lines in *Mycoplasma pneumonia* [31].

**Figure 5 diagnostics-13-01612-f005:**
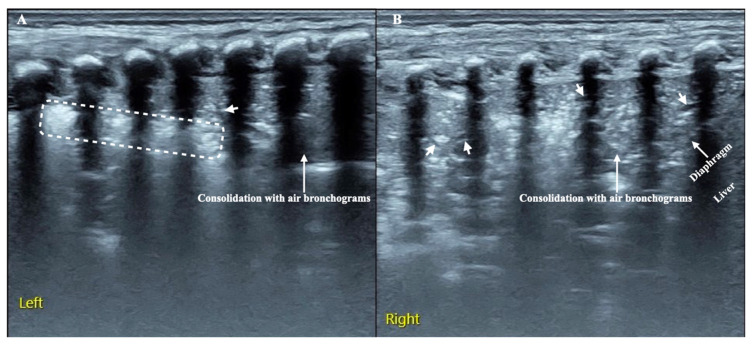
Consolidation with air bronchogram in fungal pneumonia [37].

**Figure 6 diagnostics-13-01612-f006:**
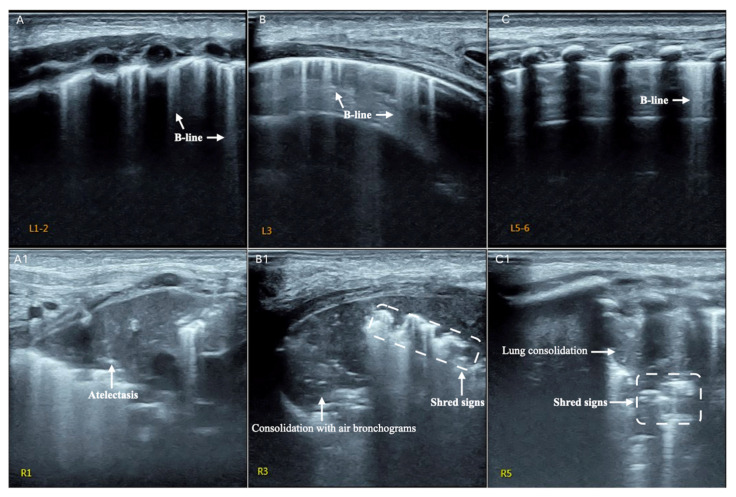
B-lines (**A**–**C**), Atelectasias, consolidation with air bronchogram, shred signs in fungal pneumonioa [37].

**Figure 7 diagnostics-13-01612-f007:**
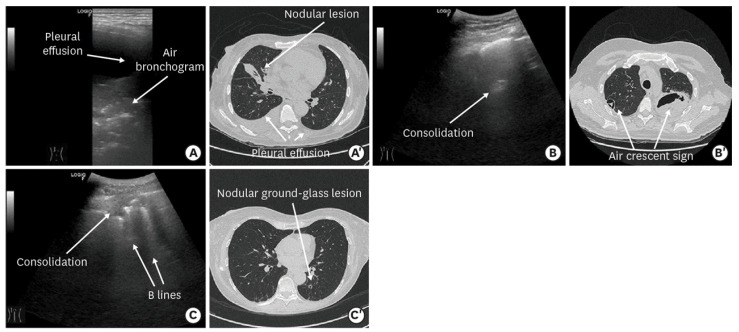
Comparison between LUS images and CT images in fungal pneumonia [40].

**Figure 8 diagnostics-13-01612-f008:**
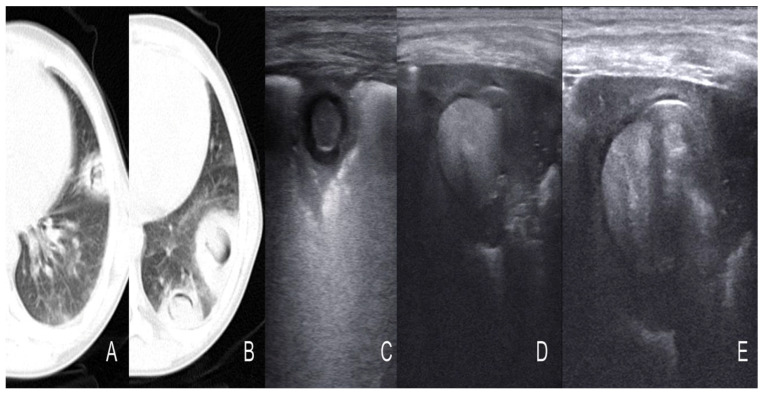
(**A**,**B**) Multiple nodular lesions with peripheral ground-glass opacity or halo sign, reverse halo sign, and air crescent sign. (**C**) The target lesions; a hyper-echoic central nodule with a hypo-echoic rim. (**D**) Hyper-echoic nodule (fungus ball) within consolidaion. (**E**) Hyper-echoic fungus ball with air exension at the peripheral hup-echoic rim [36].

**Figure 9 diagnostics-13-01612-f009:**
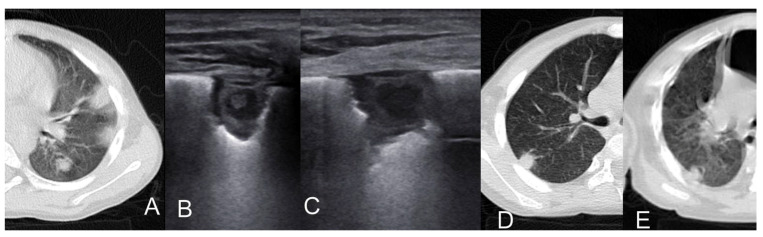
(**A**) Multiple nodular lesions with ground-glass opacity or halo sign, without air crescent sign. Two target lesions in ultrasound. (**B**) Hyper-echoic central nodule with a hypo-echoic rim. Figure (**C**) Hypo-echoic center with a hyper-echoic rim. (**D**,**E**) A nodular lesion of the right lung that has developed and air crescent sign in control HRCT [36].

**Figure 10 diagnostics-13-01612-f010:**
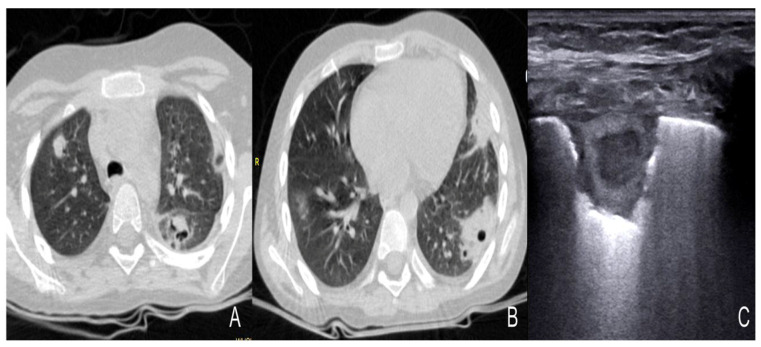
(**A**,**B**) Multiple nodular lesions with cavitation. (**C**) The target lesion in ultrasound has a Hypo-echoic center with a hyper-echoic rim [36].

**Figure 11 diagnostics-13-01612-f011:**
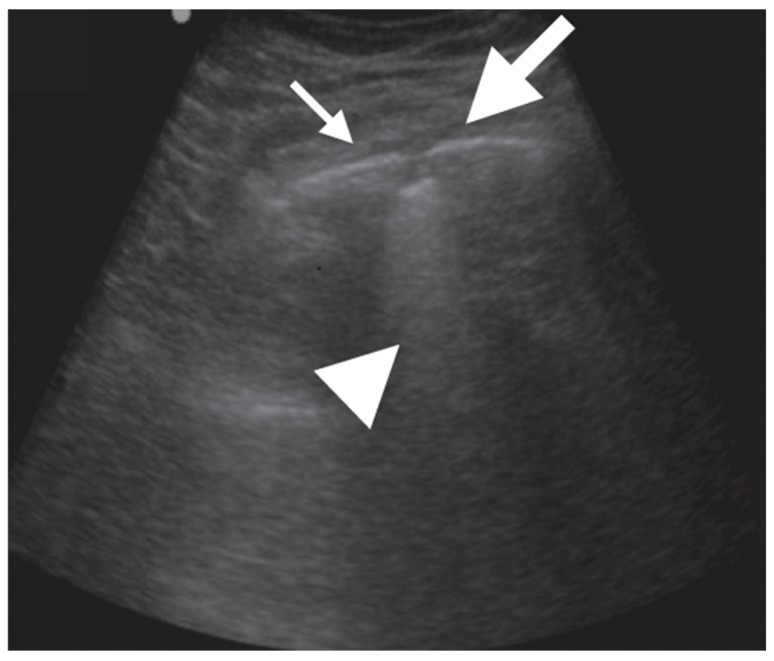
Subpleural consolidation (thick arrow), characterized by subpleural hypo-echoic region < 1 × 1 cm, with distinct borders and trailing artifact (arrowhead), next to the normal white pleural line (thin arrow) [44].

**Figure 12 diagnostics-13-01612-f012:**
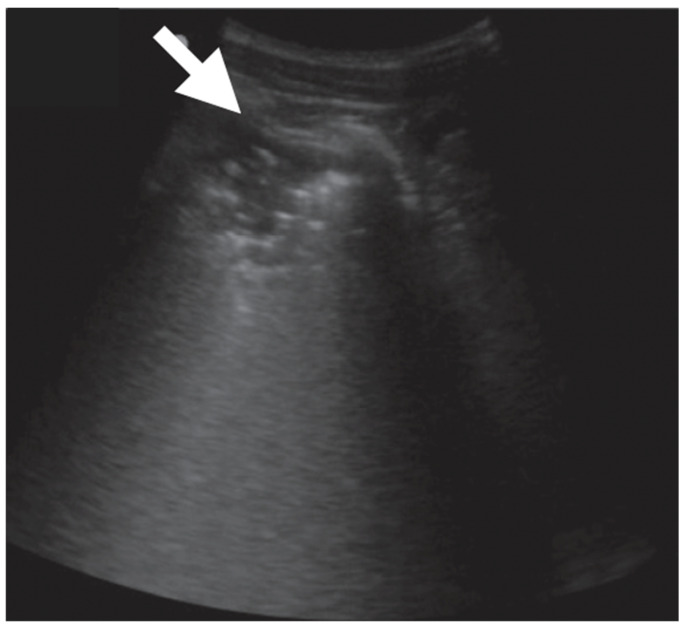
Consolidation (thick arrow), characterized by echo-poor region > 1 × 1 cm, with air bronchograms [44].

**Figure 13 diagnostics-13-01612-f013:**
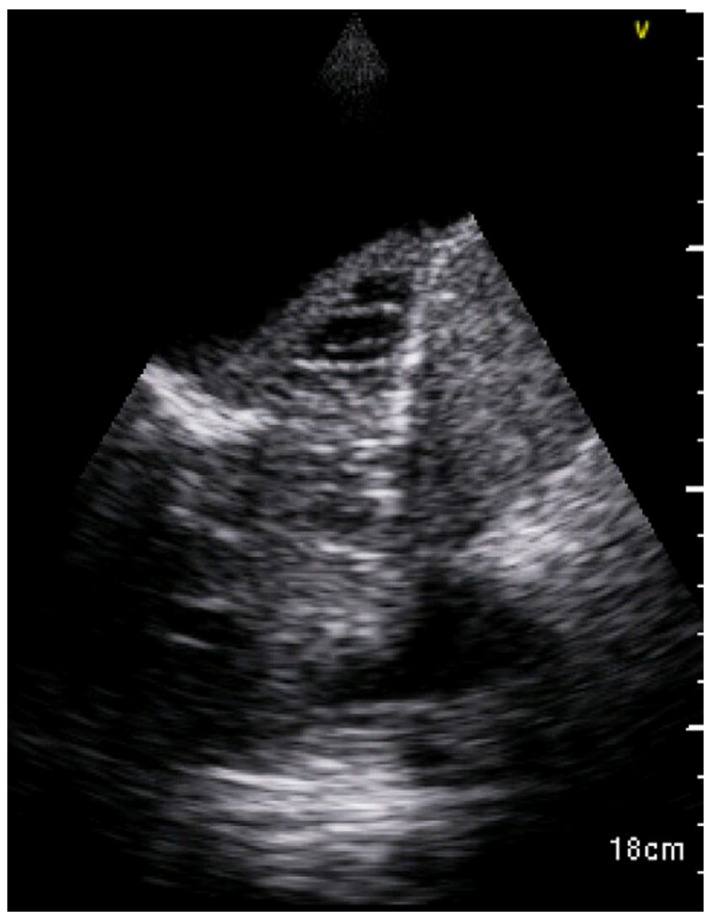
Cavitation in mycobacterial pneumonia [45].

**Table 1 diagnostics-13-01612-t001:** Definitions of the main lung ultrasound findings.

	Definition
Consolidation	Area in which lung tissue is de-aerated with density similar to parenchymal tissues [23]
Atelectasis	Type of consolidation shown as hyperechogenic tissue structure visualized as solid parenchyma with static air bronchogram [11,24]
Cavitation	Solid, hypoechoic, heterogeneous lesions with sharp lobulated margins [11,25]
Pleural effusion	Hypo- or anechogenic structure, delineated by the chest wall and the diaphragm [11,26]
B-lines	Vertical reverberation artefacts from the pleural line to the edge of the scree; laserlike, vertical hyperechogenic artefacts synchronized with pleural line [11,23,27]
Pleural irregularities	Reduction or interruption of pleural line [11,28]
Sub-pleural nodes/granularities	Hyperechogenic subcentimetric granularities or consolidation under the pleural line [29,30]

**Table 2 diagnostics-13-01612-t002:** Lung ultrasonographic findings in bacterial intracellular pathogens infection.

	N. of Patients	Age	Pleural Effusion	Consolidation		B-Lines			Atelectasias
				Total	>1–1.5 cm	Total	Scattered	Confluent	
Li, 2021 [31]	30	Mean 9 yrs	16	30 (100%)	NS	30	30	30	18
			(53%)			(100%)	(100%)	(100%)	(60%)
Tripaldi, 2021 [32]	40	Mean 4 yrs	6	34	31	13	NS	NS	NS
			(15%)	(85%)	(77%)	(32%)			
Buonsenso, 2022 [33]	43	Mean 7 yrs	7	38	19	28	28	0	26
			(16%)	(88%)	(44%)	(65%)	(65%)		(60%)
Liu G., 2022 [34]	161	Median 4 yrs	48	136	NS	161	3	154	NS
			(29%)	(84%)		(100%)	(1%)	(95%)	
Total	274		77	238 (86%)	50	232	61	184	44
			(28%)		(60%)	(84%)	(26%)	(79%)	(60%)

All patients with *M. pnuemonia* infectin; Buonsenso 2022 included also *Chlamydia pneumonia* infection. NS: the exact number is not specified in the study, thus excluded from calculation of total (%).

**Table 3 diagnostics-13-01612-t003:** Lung ultrasonographic findings in fungal invasive pneumonia.

	N. of Patients	Age	Microorganism	Consolidation	Atelectasis	Cavitation	Hyper– Echoic Nodule with Hypo– Echoic Rim	Hypo– Echoic Nodule with Hyper– Echoic Rim	Pleural Effusion	B–lines
Children										
Trinavarat 2012 [35]	1	6 weeks	Aspergillus	1		1				
				(100%)		(100%)				
Alamdara, 2021 [36]	6	5-11 yrs	1 Mucurmicosis, 5 Aspegillus	5		2	2	4		
				(83%)		(33%)	(33%)	(66%)		
Liu J., 2022 [37]	7	Premature newborns	5 C. albicans, 1 C. parapsilosis, 1 Aspergillus	7	2				2	7
				(100%)	(33%)				(33%)	(100%)
Total	14			13	2	3	2	4	2 (14%)	7
				(93%)	(14%)	(21%)	(14%)	(28%)		(50%)
Adults										
Tikkakoski, 1995 [38]	4	49-79 yrs	4 Aspergillus (1 A.niger, 2 A.fumigatus)	4		4			1 (25%)	
				(100%)		(100%)				
Grabala, 2017 [39]	1	41 yrs	1 Mucurmicosis	1						1 (100%)
				(100%)						
Greco, 2019 [40]	10	Mean 44 yrs	Not specified	10	4				4 (40%)	8
				(100%)	(40%)					(80%)
Ruby, 2021 [41]	1	45 yr	1 Aspergillus	1		1	1			
				(100%)		(100%)	(100%)			
Total	16			16	4	5	1		5 (31%)	9
				(100%)	(25%)	(31%)	(6%)			(56%)

**Table 4 diagnostics-13-01612-t004:** Ultrasonographic findings in *Mycobacterium tuberculosis*.

	N. of Patients	Mean Age	Pleural Effusion	Pleural Irregularities	Consolidation	Cavitation	Subpleural Nodes	Subpleural Granularities	B-Lines
Children									
Heuvelings, 2019 [42]	40	2 yrs	12 (30%)	31	22	0	0	0	13
				(78%)	(55%)				(33%)
Adults									
Hunter, 2016 [30] *	10	33 yrs median	0	0	0	0	0	10	10
								(100%)	(100%)
Agostinis, 2017 [29]	60	32 yrs	11	NS	28	3	58	4	NS
			(18%)		(46%)	(5%)	(98%)	(6%)	
Montuori, 2019 [43]	51	34 yrs	10	37	40	9	37	0	0
			(20%)	(73%)	(77%)	(30%)	(73%)		
Fentress, 2020 [44]	51	34 yrs	4	NS	41	3	41	0	20
			(7%)		(80%)	(6%)	(80%)		(39%)
Cocco, 2022 [45]	1	36 yrs	1 (100%)	0	1	1	0	0	0
					(100%)	(100%)			
Total (adults)	173		26	37	110	16	136	14	30
			(15%)	(59%)	(63%)	(9%)	(78%)	(8%)	(26%)

* Only cases of miliary tuberculosis included. NS: the exact number is not specified in the study, thus excluded from calculation of total (%).

## Data Availability

No new data were created.

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
