# Peer review of "The Role of Ultrasound in the Diagnosis of Pulmonary Infection Caused by Intracellular, Fungal Pathogens and Mycobacteria: A Systematic Review"

_diagnostics, 2023, doi:10.3390/diagnostics13091612_

Round 1

Reviewer 1 Report

The authors have presented a review on the role of ultrasound in the diagnosis of pulmonary infection caused by intracellular, fungal pathogens and mycobacteria. The review will be of importance to the relevant community. 

How the authors can say that this review is PRISMA compliant?

Fig 1 is misleading and does not provide the information correctly. Please modify and correct the same. 

The conclusions drawn from the study are not clear and conclusive. Please describe, clarify and substantiate with evident (drawing conclusion from particular paper)

Author Response

Dear reviewer,

thank you for the comments and suggestions.

We used an evidence-based algorithmic approach, Preferred Reporting Items for Systematic Reviews and Meta-Analyses (PRISMA) and we have modified the lacking data, adding more information about the study selection and data extraction. Changes were added in the text according to your suggestions. 

Figure 1 describes PRISMA: it was modified according to your suggestions and was insert in the right place of the text.

Conclusions were clarified describing LUS images in the different infections, underling the most frequent findings in each type of infection.

Reviewer 2 Report

I would like to thank the handling editor for giving me the opportunity to review the manuscript entitled “The role of ultrasound in the diagnosis of pulmonary infection caused by intracellular, fungal pathogens and mycobacteria: a systematic review” by Meli and colleagues submitted to Diagnostics for consideration of publication. I also want to congratulate the authors for their work, which represents a systematic review investigating the use of ultrasonography for the diagnosis of lung infections caused by intracellular bacteria, fungal microorganisms, and mycobacteria. The review was reported according to the Preferred Reporting Items for Systematic Reviews and Meta-Analyses (PRISMA) statement. After searching the PubMed, Embase, Google Scholar, and Cochrane electronic databases, the authors identified 695 studies in total. Of those, 17 were included in the final qualitative analysis. These studies reported 227 patients with M. pneumoniae, 30 with fungal infection, and 213 with pulmonary tuberculosis. Most of the studies on M. pneumoniae in children found a specific pattern on lung ultrasound that was mainly characterised by consolidated areas associated with diffuse B lines. The typical features in tuberculosis consisted of consolidation and small subpleural nodes. Regarding ultrasound findings of fungal lung infection, only one study reported specific patterns (i.e., “halo sign” or “reverse halo sign”). Therefore, the authors conclude that lung ultrasound appears to be a promising point-of-care tool for differentiating atypical pneumonia or tuberculosis from common bacterial pulmonary infections. They recognise, however, that the use of ultrasound in the diagnosis of fungal lung disease is still at an early stage of exploration, and large trials are warranted to further elucidate its role in this setting. This is an interesting and informative review that has the potential to be a valuable addition to the pertinent literature.

The submitted paper is generally well-composed, yet there is scope for enhancement. The introduction effectively establishes the necessary context, even for readers who possess limited knowledge about the subject matter. Furthermore, the findings are presented clearly with pertinent tables and are discussed in relation to recent literature. The conclusions are drawn from the systematic review's results and bear clinical relevance. I would like to offer several suggestions that, in my view, could contribute to the overall quality of the manuscript:

·         In the Abstract, the authors report that they “included 17 studies, reporting a total of 227 patients with M. pneumoniae, 30 with fungal infection and 213 with pulmonary tuberculosis”. However, the number of included patients appears to be different in the Results, as presented in Tables 2, 3, and 4 of the manuscript. The authors may wish explaining this discrepancy.

·         In the Methods section, the authors might consider detailing the date on which each database was last searched.

·         Instead of merely listing the search terms, it could be beneficial for the authors to present the complete search strategies employed for all the databases that were explored.

·         The authors might want to delineate the methods utilized to determine whether a study fulfilled the review's inclusion criteria and the techniques employed to gather data from reports (e.g., possible use of automation tools).

·         The authors should consider elaborating on the methods employed to assess the risk of bias in the included studies. This should encompass details of the tool used and the number of reviewers who assessed each study. Subsequently, they may wish to include relevant risk of bias assessments for each included study in the Results section.

·         In the PRISMA flowchart, the authors might consider indicating the number of excluded manuscripts along with the reasons for exclusion. Additionally, they should clarify the meaning of the term "useless data."

·         Finally, the authors may wish to address any limitations inherent in the review processes employed.

Author Response

Dear reviewer,

thank you for the comments and suggestions.

We have changed the numeric discrepancy among the Abstract, the text and the tables.

We modified Material and Methods, as you suggested: the changes were added and highlighted in the text.

We have considered the risk of bias of the 17 studies included; it is low for the patient selection in most of cases. It is difficult to evaluate index test, reference standard, flow and timing domains because LUS is a new diagnostic tool and few data are available about its use in intracellular, fungal pathogens and mycobacterium lung infections.

In PRISMA flow-chart, we added the reason for exclusion manuscripts, and we explained the meaning of ‘useless data’.

Dear reviewer,

thank you for the comments and suggestions.

We have changed the numeric discrepancy among the Abstract, the text and the tables.

We modified Material and Methods, as you suggested: the changes were added and highlighted in the text.

We have considered the risk of bias of the 17 studies included; it is low for the patient selection in most of cases. It is difficult to evaluate index test, reference standard, flow and timing domains because LUS is a new diagnostic tool and few data are available about its use in intracellular, fungal pathogens and mycobacterium lung infections.

In PRISMA flow-chart, we added the reason for exclusion manuscripts, and we explained the meaning of ‘useless data’.

Reviewer 3 Report

This authors reviewed 17 studies on the most common LUS patterns in infections caused by intracellular, fungal pathogens or mycobacteria. The results from those studies show that LUS could be used a promising point-of-care tool to show patterns of atypical pneumonia and TB which seem different from patterns characterizing common bacterial infection.

Major comments:

1. According to Table 4, cases in ref [30] are special compared to other research, then what’s the reason it is still necessary to be included in the review?

2. It’s better to clearly list the 17 studies that were finally chosen by the authors in a table with some basic information of each study.

Minor comments:

1. In Figure 7 please indicate in the caption why subfigures ABC and DEF are listed separately if there’s a reason to do so.

2. In Figure 9, please align the labels A~E in each subfigure

3. Please try to adjust the Tables, some of them are out of the page boundary, e.g., Table 2~4.

Author Response

Dear reviewer,

thank you for the comments and suggestions.

We have maintained ref [30] because the study describes 10 of 173 patients with mycobacterial pneumonia.

We decided to realize 3 different tables, each one for specific infection: atypical bacterial, fungal and mycobacterial pneumonias.

We edited the tables how you suggested. The Figure 7 and 9 were reported as in the original articles.  

Round 2

Reviewer 2 Report

I would like to thank the authors for considering my suggestions and revising their manuscript accordingly.